# Interaction Among Influenza Viruses A/H1N1, A/H3N2, and B in Japan

**DOI:** 10.3390/ijerph16214179

**Published:** 2019-10-29

**Authors:** Ayako Suzuki, Kenji Mizumoto, Andrei R. Akhmetzhanov, Hiroshi Nishiura

**Affiliations:** 1Graduate School of Medicine, Hokkaido University, Kita 15-Jo Nishi 7-Chome, Kita-ku, Sapporo-shi, Hokkaido 060-8638, Japan; aya_suzu@med.hokudai.ac.jp (A.S.); akhmetzhanov@med.hokudai.ac.jp (A.R.A.); 2CREST, Japan Science and Technology Agency, Honcho 4-1-8, Kawaguchi, Saitama 332-0012, Japan; 3Graduate School of Advanced Integrated Studies in Human Survivability, Kyoto University Yoshida-Nakaadachi-cho, Sakyo-ku, Kyoto 606-8306, Japan; mizumoto.kenji.5a@kyoto-u.ac.jp

**Keywords:** influenza, viral interference, epidemics, statistical model, epidemiology

## Abstract

Seasonal influenza epidemics occur each winter season in temperate zones, involving up to 650,000 deaths each year globally. A published study demonstrated that the circulation of one influenza virus type during early influenza season in the United States interferes with the activity of other influenza virus types. However, this finding has yet to be validated in other settings. In the present work, we investigated the interaction among seasonal influenza viruses (A/H1N1, A/H3N2 and B) in Japan. Sentinel and virus surveillance data were used to estimate the type-specific incidence from 2010 to 2019, and statistical correlations among the type-specific incidence were investigated. We identified significant negative correlations between incidence of the dominant virus and the complementary incidence. When correlation was identified during the course of an epidemic, a linear regression model accurately predicted the epidemic size of a particular virus type before the epidemic peak. The peak of influenza type B took place later in the season than that of influenza A, although the epidemic peaks of influenza A/H1N1 and A/H3N2 nearly coincided. Given the interaction among different influenza viruses, underlying mechanisms including age and spatial dependence should be explored in future.

## 1. Introduction

Influenza is an acute viral respiratory disease that is involved in approximately one billion infections worldwide [1]. Symptoms of influenza are often self-limiting, with recovery occurring within 7 days. Typical clinical signs and symptoms of influenza include sudden onset of fever or upper respiratory symptoms such as sore throat, cough, and runny nose [2]. However, influenza infection in high-risk individuals, including older people with underlying comorbidities, can result in severe complications, which lead to approximately 650,000 deaths each year [1,3]. There are four types of influenza viruses, including A, B, C, and D [4], among which type A and B viruses continuously circulate and cause annual epidemics in humans. Currently circulating virus types include influenza A/H1N1, influenza A/H3N2, and influenza B (which is further subdivided into B/Victoria and B/Yamagata) [5].

Influenza vaccination is the most effective way to reduce the risk of infection or severe forms of infection [6]. However, the effectiveness of existing vaccines is limited because antigenic drift in influenza viruses over time requires annual immunization and that vaccines are updated frequently [7]. To develop an effective vaccine strategy, it is vital to predict the likely predominant virus type (A/H1N1, A/H3N2, or B) and its antigenic characteristics [8]. Moreover, co-circulation of virus types does not take place in an independent manner. Sonoguchi et al. [9] reported that Japanese high school students who were infected with influenza had a lower risk of subsequent infection with A/H1N1 during the period of A/H1N1 and A/H3N2 co-circulation, indicating short-term cross-subtype protection induced by the initial influenza infection. In observational studies, Skowronski et al. [10] found that people who received influenza vaccination during 2008–2009 were more likely to have symptoms owing to influenza A/H1N1pdm in 2009 than those who did not receive influenza vaccination during 2008–2009. Cowling et al. [11] reported that the risk of infection with influenza A/H1N1pdm in 2009 among people who were infected with seasonal influenza A during 2008–2009 was significantly lower than that among people without seasonal influenza A infection; those authors believed that the temporary nonspecific immunity theory could explain short-term protection induced by influenza infection [12]. Several modeling studies [13,14,15,16] have applied the temporary nonspecific immunity theory to describe influenza virus dynamics and evolution, although biological evidence is limited. Goldstein et al. [17] showed that possible interference among three influenza virus types, A/H1N1, A/H3N2, and B, as a consequence of cross protection and early onset of one virus type, could lead to reduced incidence of other co-circulating influenza viruses. Additionally, those authors indicated that routine surveillance data could be used to predict the epidemic size for each influenza virus type. The interference among different virus types is also important in the context of pandemic preparedness. It is argued that the required stockpiles of antiviral drugs (e.g., oseltamivir) should be estimated, as part of preparedness for future pandemics [18]. The Japanese government has maintained additional antiviral stockpiles to prepare for possible simultaneous epidemics caused by seasonal influenza viruses. If influenza viruses do not independently cause epidemics, such arguments require additional insights into the mutual interference mechanisms.

The predictability of relative epidemic sizes for each virus type using epidemic data from early phases of the influenza season has been explored using state-specific datasets from the United States [10]. However, this has not been investigated in other settings. Data from a single country reflect climatologically-diverse dynamics [19], and surveillance data are collected from various geographic units [20]. In line with this, Japan has continuously implemented so called virus surveillance as well as sentinel influenza surveillance [21]. Virus surveillance can detect virus types and subtypes over time whereas sentinel surveillance collects notifications of influenza diagnoses from physicians at designated health care facilities. In Japan, most notified cases (but not cases of influenza-like illness) undergo rapid diagnostic testing (RDT). This detects influenza virus (with virus typing information) from nasopharyngeal swabs on bedside, including at outpatient settings, and those with positive results are notified. Even excluding the latest pandemic year (2009), the continual surveillance system provides an opportunity to examine the interaction among influenza virus types and subtypes.

In the present study, we aimed to investigate the interaction between influenza A/H1N1, A/H3N2, and B in Japan with respect to the size and timing of epidemics.

## 2. Materials and Methods

### 2.1. Epidemiological Data

To estimate the type-specific influenza incidence, the present study analyzed two datasets: sentinel surveillance data and laboratory-based virus surveillance data, across all 47 prefectures of Japan. Two sets of data were combined, because sentinel surveillance informs the overall incidence of influenza, while virus surveillance data with typing information are based on a small portion of confirmed cases. Multiplying the percentage of each virus type with sentinel reports of influenza notifications, we obtain type-specific proxy of influenza incidence. Adhering to the Infectious Disease Law and under the National Epidemiological Surveillance for Infectious Diseases (NESID) program, influenza cases are notified to local health centers via health care facilities. The weekly number of influenza cases is reported from approximately 5000 sentinel health care facilities in Japan [22]. Virus surveillance is independently conducted, reporting the weekly frequency of virus types (i.e., A/H1N1pdm, A/H1N1, A/H3N2, B/Victoria, and B/Yamagata) using specimens collected from approximately 500 health care facilities sampled from sentinel health care facilities [23]. Untyped A viruses are not reported. Diagnostic criteria for notified influenza cases are either a positive RDT for influenza and/or the presence of symptoms of influenza-like illness (ILI), defined as satisfying all of the following: (1) sudden onset of illness, (2) high-grade fever, (3) upper respiratory symptoms, and (4) systemic symptoms including general fatigue [24]. As briefly described in the introduction, physicians in Japan regularly use RDT in patients with suspected influenza infection, because the use of RDT is financially covered by health insurance under their universal health coverage. Thus, most notified cases of influenza are considered to represent the true influenza diagnoses using RDT, mirroring the influenza incidence in a more appropriate manner than ILI-based surveillance in other countries.

Here, we used a proxy measure of the type-specific incidence (A/H1N1, A/H3N2, and B), calculating the incidence as the product of the weekly number of reported influenza cases and the proportion of virus type identification for the week among the three virus types. Throughout this article, we define the index virus as the dominant virus type in an epidemic season. When choosing one virus type as the index, we defined the sum of the incidence of the other two virus types as the complementary incidence.

Japanese influenza surveillance monitors influenza trends from calendar week 36 to calendar week 35 (from October to the end of September in the subsequent year). Epidemiological week 1 corresponds to calendar week 36. Because the epidemic size of A/H1N1 in 2009 was exceptionally high owing to the A/H1N1 pandemic, and our aim in this study was to investigate the interference among seasonal influenza viruses, we analyzed data from calendar week 36 in 2010 to calendar week 35 in 2019.

### 2.2. Statistical Analysis

We used existing methods introduced by Goldstein et al. [17] in a study conducted in the United States. To assess the interaction of epidemic sizes among the three influenza viruses, we used the Spearman rank-order method to explore the correlation between the epidemic size and cumulative complementary incidence, up to a specific week. We investigated not only the dynamic interactions but also the correlation of epidemic sizes among the different virus types. We examined the Spearman rank correlation between the cumulative incidence of the index virus and that of the complementary viruses.

For dynamic assessment of the interaction, we first identify the earliest week at which the cumulative incidence of the index virus (i.e., dominant virus type) becomes predictable. Subsequently, we show that the cumulative incidence of complementary virus types can also be predicted based on a negative correlation with the index virus. To accomplish the first task, we assumed that the epidemic size for the index virus in each season *Y* can be estimated using the linear regression model Y=βX at the time point (epidemiological week *s*) when one of the following conditions is first met: the cumulative incidence of the index virus in the last 5 weeks reaches the threshold *h* or the cumulative complementary incidence reaches the threshold *h*_c_. Here, covariate *X* is linked to the growth of the incidence of the index virus before epidemiological week *s*, expressed as
(1)$$X=\frac{I\left(s\right)+I\left(s-1\right)}{max\left(h,\sum_{a=0}^{a=4}I(s-a,)\right)}$$
where *I*(*s*) indicates that the incidence for the index virus in week *s*. Here the latest two-week data are summed in the numerator for smoothing purpose, and the denominator is the sum over the latest five weeks to compare against the numerator and understand the growth in the recent time. As the size and pattern of epidemics are quite different among the three influenza virus types, thresholds *h* and *h*_c_ were chosen for each virus type. To determine the thresholds *h* and *h*_c_, we calculated the residual standard error (RSE) for the epidemic size using the equation:(2)$$RSE=\sqrt{\frac{{\left(Y_{obs}-Y_{pred}\right)}^{2}}{DF}}$$
where DF denotes the number of the degrees of freedom, calculated as the number of influenza seasons minus 1. *h* and *h*_c_ for each virus type were chosen from the combinations of *h* and *h*_c_ that gave the smallest RSE. Namely, abovementioned thresholds represent the most reasonable early week at which the cumulative incidence has become predictable.

To investigate the interaction among the three influenza viruses (A/H1N1, A/H3N2, and B) with respect to the peak incidence timing, we examined differences in the weeks at which incidence reached the peak (i.e., the maximum value during a single influenza season) among the three influenza virus types.

### 2.3. Ethical Considerations

This study used publicly available data that were completely anonymized. Therefore, ethical approval was not required for the current study.

## 3. Results

Epidemic curves in Figure 1 show the temporal distribution of the weekly incidence of three influenza viruses (A/H1N1, A/H3N2 B) from 2010 to 2019. The epidemic curve for each virus type shows the various shapes each year, and there is no specific multi-year trend for any single virus type. Among the nine seasons, A/H3N2 predominated in six seasons, A/H1N1 was predominant in three seasons and B was predominant during one season (2017–2018).

Figure 2 shows the temporal distribution of the weekly incidence for influenza A/H3N2. Vertical lines indicate the week *s* of each season when the prediction of the epidemic size took place. The distribution of the weekly incidence of influenza A/H1N1 and B and the prediction timings are shown in the Appendix A. Using the optimization process with equation (2), selected thresholds for influenza A/H1N1, A/H3N2, and B were *h* = 250,000 and *h*_c_ = 570,000, *h* = 480,000 and *h*_c_ = 700,000, and *h* = 310,000 and *h*_c_ = 810,000 cases, respectively. Appendix A shows the heatmaps of the RSE for various combinations of thresholds *h* and *h*_c_. The index virus first reached its own threshold value in four seasons (2010–2011, 2013–2014, 2015–2016, and 2018–2019) for influenza A/H1N, in five seasons (2011–2012, 2012–2013, 2014–2015, 2016–2017, and 2018–2019) for influenza A/H3N2, and in two seasons (2015–2016 and 2017–2018) for influenza B. In these seasons, we observed large epidemics of the index virus type. Prediction of the epidemic size of the season was mostly attained in or before the same week as the epidemic peak except in the 2012–2013 season for influenza A/H1N1 and the 2012–2013 and 2013–2014 seasons for influenza A/H3N2 and B. Details of the prediction timing and the peak timing of the epidemic are shown in Appendix A.

Figure 3A–C show the Spearman rank correlation between the epidemic size of the index virus and the cumulative complementary incidence, up to epidemiological weeks 22–24 from 2010 to 2019. Although some correlations were marginally- or non-significant, negative correlation was identified in this dynamic assessment of the relationship. Figure 3D–F show the Spearman rank correlations between the epidemic size for the index virus and the cumulative complementary viruses from 2010 to 2019. Negative correlation was also observed between the epidemic size of the index virus and the cumulative complementary viruses.

Figure 4A–C depict the predicted epidemic size using the proposed simplistic linear model against the observed data of influenza A/H1N1, A/H3N2, and B from 2010 to 2019. Detailed information of the regression model for each virus type is summarized in Table 1.

Figure 5 illustrates the interrelationship of timings at which the peak incidence is observed among the three virus types. When comparing influenza type A viruses (i.e., A/H1N1 vs. A/H3N2), the mean difference between the two peak weeks was estimated at 0.11 weeks. Comparing the timing of peak weeks between influenza type A and B viruses, the mean differences were 2.22 and 2.11 weeks between influenza type B and A/H1N1 and influenza type B and A/H3N2, respectively. Standard deviations ranged from 2.51 to 4.18 weeks.

## 4. Discussion

In the present study, we explored seasonal influenza epidemic data in Japan from 2010 to 2019, combining sentinel and virus surveillance data for a total of nine epidemic seasons. We investigated statistical interactions between different influenza virus types during the course of and at the end of epidemics. We revealed that the cumulative incidence of the index virus was negatively correlated with both the dynamic cumulative incidence of the complementary viruses around the peak week and the entire epidemic size of the complementary viruses. Specifying the index virus type in a single influenza season, this index virus was found to be predictable using a simple linear regression model, mostly before observing the epidemic peak. The timing of the epidemic peak was also examined across different types of influenza virus. We did not identify interacting timing between influenza A/H1N1 and A/H3N2, and the peak timing was likely different when comparing influenza type A and B viruses (type A virus tended to precede type B).

The present study verifies that seasonal influenza virus types interact with each other, and negative correlations are seen around the time at which the predominant type becomes predictable. Our findings indicate that it is vital to account for the introduction of viruses during real-time prediction of any type of seasonal influenza because the incidences interfere with each other. As long as different influenza virus types continue to co-circulate, it must be remembered that the epidemic size is not determined according to transmissibility and herd immunity alone. Goldstein et al. [17] first conducted an epidemiological analysis of interference among co-circulating influenza viruses in a state-specific setting of the United States [5]. The present study adds to the literature regarding the interference among different influenza virus types, identifiable using prefecture-specific datasets in Japan, another temperate zone country.

In observing negative correlations with incidence between index and complementary viruses, we also showed that the underlying mechanisms of interactions might differ for the interaction between influenza A/H3N2 and A/H1N1 and that between influenza type A and B viruses. As for the latter interaction, the consistent pattern of influenza type A virus predominance, followed by predominance of influenza type B virus, is indicative of differential seasonal preference or other mechanisms that have yet to be identified. In relation to stockpiling of oseltamivir in planning for potential simultaneous epidemics of different virus types, our findings indicate that the epidemic peak of pandemic virus may coincide with that of another influenza type A virus, although the epidemic peak of pandemic virus may not coincide with that of influenza type B virus. Nevertheless, it must be remembered that the population was largely susceptible to historically emerged pandemic viruses in the 20th and 21st centuries at the time of their emergence and perhaps more susceptible in comparison with seasonal influenza virus of the same subtype. Thus, the interference between pandemic and seasonal influenza viruses (type A) could be more abrupt than the interactions among seasonal viruses. In fact, influenza A/H3N2 was seldom observed during the course of the 2009 pandemic.

Four limitations of this study must be discussed. First, there was no systematic sampling scheme used for virus surveillance data. For instance, severe cases with influenza A/H3N2 might have been more frequently sampled at hospitals than influenza A/H1N1; thus, the examined empirical data cannot reveal the full picture of all infected individuals. Second, vaccination was underway in each season, although only older people with underlying comorbidities receive routine vaccination against influenza in Japan. Homologous vaccines could have modified our results. Third, various heterogeneities were ignored, including spatial heterogeneity with respect to the predominant virus type and age specificity with respect to acquired immunity to a specific virus type. Fourth, whereas we epidemiologically identified a strong signature of interactions among different types of influenza viruses, the exact mechanisms behind our observations have yet to be elucidated.

Despite these limitations, we believe that the present study sufficiently characterizes interacting influenza virus type structures, using data from 47 prefectures during nine influenza seasons in Japan. Such interaction among seasonal influenza viruses allowed us to predict the size of influenza epidemics during the early phase of the season. Our findings provide useful insight into influenza forecasting and preparedness for future influenza pandemics.

## 5. Conclusions

We examined the interaction among seasonal influenza viruses (influenza A/H1N1, A/H3N2, and B) using sentinel and virus surveillance data in Japan. Co-circulating influenza viruses interact with each other as far as the size and peak timing of the epidemic, indicating that epidemic dynamics are not determined by transmissibility and herd immunity alone.

## Figures and Tables

**Figure 1 ijerph-16-04179-f001:**
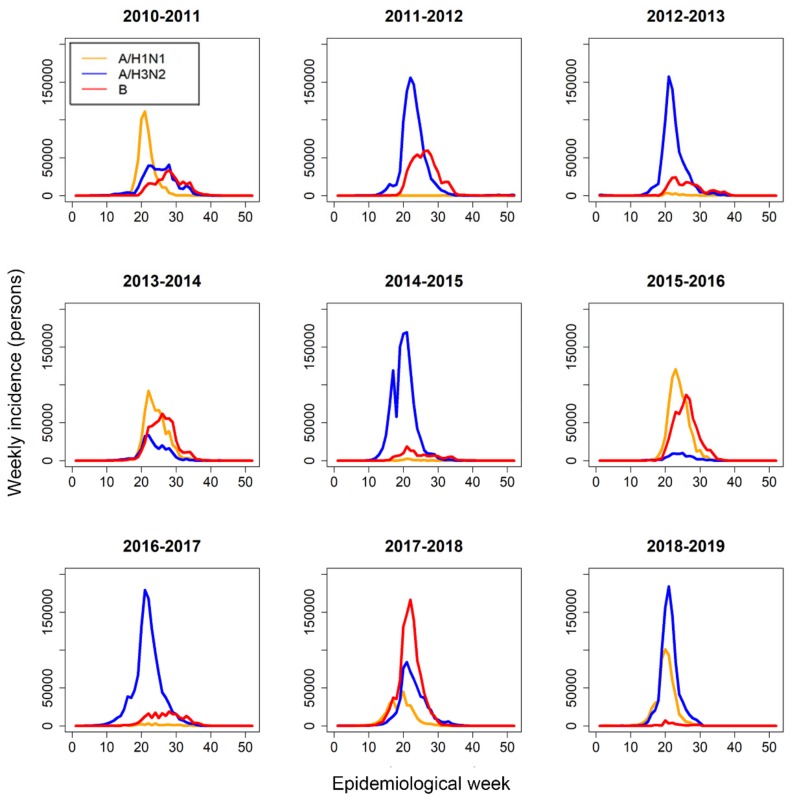
Temporal distribution of the weekly incidence of three influenza virus types, A/H1N1 (orange), A/H3N2 (blue), and B (red) from 2010 to 2019. Epidemiological week 1 corresponds to calendar week 36 in each year.

**Figure 2 ijerph-16-04179-f002:**
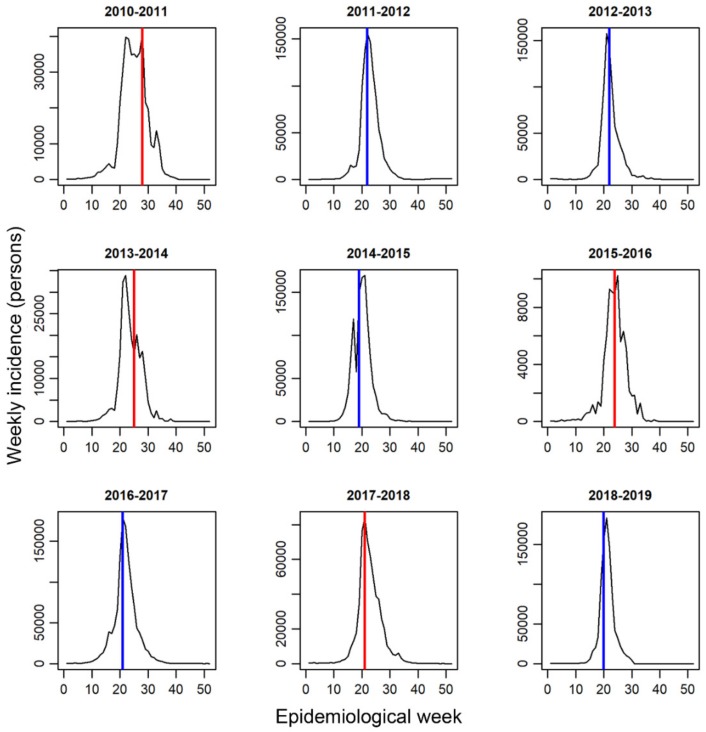
Temporal distribution of the weekly incidence proxies for influenza A/H3N2 from 2010 to 2019. Epidemic week 1 corresponds to calendar week 36. Blue lines indicate the week at which the prediction of the epidemic size took place in each season, corresponding to the time at which the sum of the incidence of influenza A/H3N2 in the past 5 weeks exceeds the statistically chosen threshold *h* = 480,000 cases. Red lines indicate the prediction timing, corresponding to the time at which the complementary cumulative incidence from week 1 exceeds the chosen threshold *h_c_* = 700,000 cases.

**Figure 3 ijerph-16-04179-f003:**
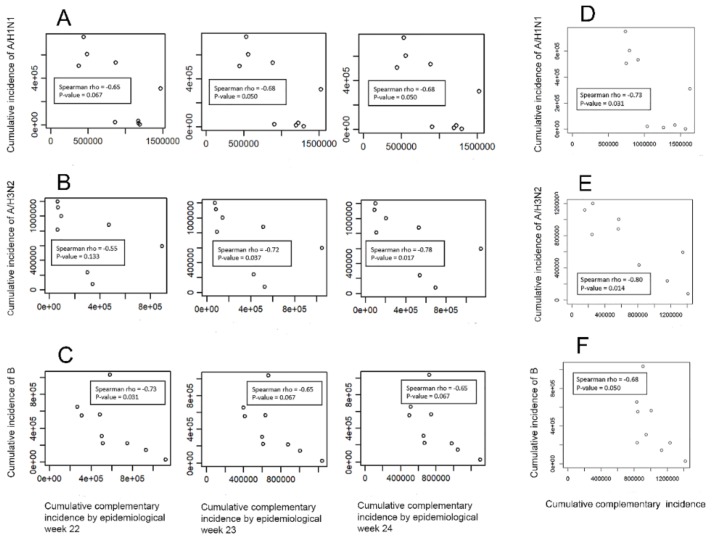
Dynamic and cumulative correlations among different influenza viruses. (**A**) Correlation between the cumulative incidence of influenza A/H1N1 and the cumulative incidence of the complementary viruses by epidemiological weeks 22–24. (**B**) Correlation between the cumulative incidence of influenza A/H3N2 and the cumulative incidence of the complementary viruses by epidemiological weeks 22–24. (**C**) Correlation between the cumulative incidence of influenza B and cumulative incidence of the complementary viruses by epidemiological weeks 22–24. (**D**) Correlation between the cumulative incidence of influenza A/H1N1 and the cumulative complementary incidence. (**E**) Correlation between the cumulative incidence of influenza A/H3N2 and the cumulative complementary incidence. (**F**) Correlation between the cumulative incidence of influenza B and the cumulative complementary incidence.

**Figure 4 ijerph-16-04179-f004:**
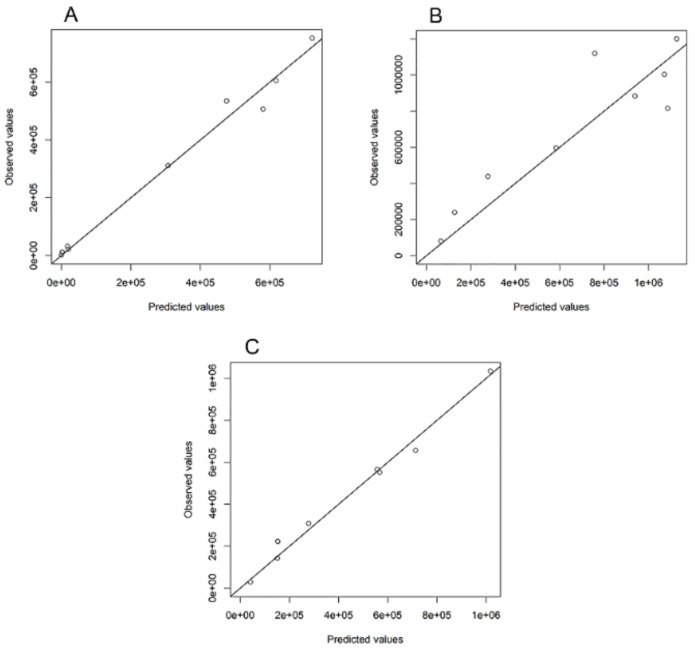
Visual assessment of prediction of epidemic sizes from 2010 to 2019. (**A**) Correlation between the predicted and observed epidemic sizes of influenza A/H1N1. The chosen thresholds were *h* = 250,000 and *h_c_* = 570,000. (**B**) Correlation between the predicted and observed epidemic sizes for influenza. The statistically chosen thresholds were *h* = 480,000 and *h_c_* = 700,000 cases. (**C**) Correlation between the predicted and observed epidemic sizes for influenza B. The statistically chosen thresholds were *h* = 310,000 and *h_c_* = 810,000 cases, respectively.

**Figure 5 ijerph-16-04179-f005:**
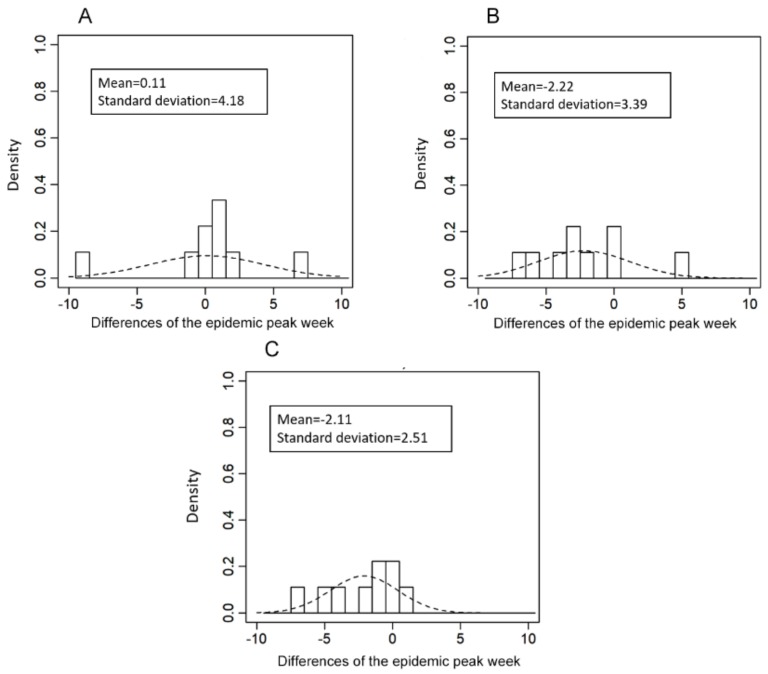
Distribution of the differences between the week of peak incidence for two influenza virus types. (**A**) Comparison of the peak timing for influenza A/H1N1 versus A/H3N2 (peak epidemic week of A/H3N2 minus that of A/H1N1). (**B**) Comparison of the peak timing for influenza B versus A/H1N1 (peak epidemic week of A/H1N1 minus that B). (**C**) Comparison of the peak timing for influenza B versus A/H3N2 (peak epidemic week of A/H3N2 minus that of B).

**Table 1 ijerph-16-04179-t001:** Prediction models of the epidemic sizes of influenza A/H1N1, A/H3N2, and B.

	Influenza A/H1N1	Influenza A/H3N2	Influenza B
	(*h* = 250,000, *h_c_* = 570,000)	(*h* = 480,000, *h_c_* = 700,000)	(*h* = 310,000, *h_c_* = 810,000)
RSE	35,864	178,905	42,923
Estimated β	977,055	1,749,322	1,651,129
Standard error	28,059	133,485	45,485
*p*-value	5.06 × 10^-1^	1.09 × 10^-1^	4.32 × 10^-1^

RSE: residual standard error.

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
