# Peer review of "Interaction Among Influenza Viruses A/H1N1, A/H3N2, and B in Japan"

_ijerph, 2019, doi:10.3390/ijerph16214179_

Round 1
Reviewer 1 Report
Well written manuscript.
Explain RDT with little more details, will be useful for readers
Line 179 change the spelling from H3M2 to H3N2
Author Response
[Point by point responses to review comments: IJERPH 620631 “Interaction among influenza viruses A/H1N1, A/H3N2, and B in Japan”]
[Responses to Reviewer 1]
Well written manuscript.
Explain RDT with little more details, will be useful for readers
>>
We additionally mention the use of nasopharyngeal swab and bedside testing in P2L76-L77.
Line 179 change the spelling from H3M2 to H3N2
>>
Corrected accordingly.
Reviewer 2 Report
Review IJERPH 620631 “Interaction among influenza viruses A/H1N1, A/H3N2, and B in Japan”
General
This manuscript describes a study attempting to characterize the epidemiologic interaction of influenza virus subtypes. I think most influenza epidemiologist would agree that influenza virus subtypes appear to interact on the population level without being aware of direct evidence of such phenomenon. Therefore, this study, even though it closely copies the one by Goldstein et al. (2011) is of interest. This is true in particular as it is conducted in different geographic setting from the Goldstein one. Yet, some aspects of the study could be improved. While the authors they clearly lay open the origin of their approach, the visual presentation of their results is almost indistinguishable from the original and thus appears to violate an unwritten law of originality. The methods are not sufficiently clearly described. I am conducting this peer review under time constraints and am confident I could understand the methods with enough effort, but I am sure most readers would appreciate some more guidance.
Specific comments
Whole paper: “index virus” is not defined, as far as I can tell. The first time the term appears in the text is on line 100/1 in the sentence “When choosing one virus type as the index, …”, but the it is not explained.Introduction, lines 91-93: While RDT is of low sensitivity, the resulting numbers will be low, but approximately proportional to the true incidence. However, it is unclear why you add influenza-like illnesses (ILI) to RDT positives to construct your incidence proxy? This is problematic as this is not necessarily proportional to the true incidence. While Goldstein uses ILI to construct his incidence proxies, he uses the %ILI to “normalize” the virus surveillance data. I suggest to use ILI as Goldstein did.
-, line 100: “Virus isolation”—virus isolation is done in cell cultures; I wonder if you are referring to virus identification by PCR?
Statistical analysis section: While the methods you use follow (extremely) closely to what Goldstein et al. did for their 2011 paper, I think the methods should be explained much better. I am not intimately familiar with that paper and, to be honest, I do not understand what you are doing and why. I have particular difficulties with the threshold calculations. Could you give a more intuitive explanation for what that is? What are the degrees of freedom? Could you give a justification for the way these quantities are calculated? After cursory browsing of the Goldstein paper, they do not seem to provide a convincing explanation.
-, lines 118/20: I do not think that expression (1) represents a growth rate of the incidence rate; should the numerator be a difference rather than a sum?
-, lines 117/8: I assume that, in the passage “the incidence of the index virus reaches the threshold h” you are referring to cumulative incidence? Please clarify.
Results, line 143/4: I find “prediction timing of the epidemic size for the season” an unfortunate and confusing choice of words.
-, lines 145-7: It appears that you, as Goldstein did, chose the thresholds “visually”?
Author Response
[Responses to Reviewer 2]
General: This manuscript describes a study attempting to characterize the epidemiologic interaction of influenza virus subtypes. I think most influenza epidemiologist would agree that influenza virus subtypes appear to interact on the population level without being aware of direct evidence of such phenomenon. Therefore, this study, even though it closely copies the one by Goldstein et al. (2011) is of interest. This is true in particular as it is conducted in different geographic setting from the Goldstein one. Yet, some aspects of the study could be improved. While the authors they clearly lay open the origin of their approach, the visual presentation of their results is almost indistinguishable from the original and thus appears to violate an unwritten law of originality. The methods are not sufficiently clearly described. I am conducting this peer review under time constraints and am confident I could understand the methods with enough effort, but I am sure most readers would appreciate some more guidance.
>>
We appreciate the reviewers’ proper assessment. We have rewritten Methods section to improve the readability (Page 3, Section 2.2).
Specific comments
Whole paper: “index virus” is not defined, as far as I can tell. The first time the term appears in the text is on line 100/1 in the sentence “When choosing one virus type as the index, …”, but the it is not explained.
>>
The definition of index virus was added to Methods (P3L107-L108).
Introduction, lines 91-93: While RDT is of low sensitivity, the resulting numbers will be low, but approximately proportional to the true incidence. However, it is unclear why you add influenza-like illnesses (ILI) to RDT positives to construct your incidence proxy? This is problematic as this is not necessarily proportional to the true incidence. While Goldstein uses ILI to construct his incidence proxies, he uses the %ILI to “normalize” the virus surveillance data. I suggest to use ILI as Goldstein did.
>>
We agree with the reviewer. The original data (i.e. data source) in Japan is the mixture of RDT positives and the so-called ILI. However, they are mostly RDT positives due to insurance coverage of RDT at outpatient settings (although the exact percentage of RDT positives out of total notifications has not been surveyed yet) (P3L101-L102). Due to RDT positives to constitute the sentinel data, we believe that our datasets better represent the influenza incidence than other countries including the United States. We explained this matter in P3L103-L104. As for the normalization of the incidence, we have also done the similar calculation, and this point was clarified in P2L86-L90.
-, line 100: “Virus isolation”—virus isolation is done in cell cultures; I wonder if you are referring to virus identification by PCR?
>>
Corrected it to be virus type identification (P3L107).
Statistical analysis section: While the methods you use follow (extremely) closely to what Goldstein et al. did for their 2011 paper, I think the methods should be explained much better. I am not intimately familiar with that paper and, to be honest, I do not understand what you are doing and why. I have particular difficulties with the threshold calculations. Could you give a more intuitive explanation for what that is? What are the degrees of freedom? Could you give a justification for the way these quantities are calculated? After cursory browsing of the Goldstein paper, they do not seem to provide a convincing explanation.
>>
We appreciate this comment very much, because this manuscript has needed this type of input from epidemiologist to better explain our methods. We have explained the strategy of dynamic assessments in words from P3L123-L126. Degree of freedom is also explained a bit more (P3L138-L139). The practical role of thresholds is also explained in P4L140-L141.
-, lines 118/20: I do not think that expression (1) represents a growth rate of the incidence rate; should the numerator be a difference rather than a sum?
>>
The numerator is the sum of incidence for the latest two weeks, while the denominator is the sum of incidence for the latest five weeks, indicating that $X$ takes a large value if the incidence is growing (this point was clarified from P3L132-L134). For smoothing purpose of bumpy epidemic curve, the sum is used rather than taking difference. To avoid confusion, the wording of “growth rate” by Goldstein et al. was corrected to be simply “growth of incidence” in the revised manuscript (P3L130).
-, lines 117/8: I assume that, in the passage “the incidence of the index virus reaches the threshold h” you are referring to cumulative incidence? Please clarify.
>>
The reviewer is correct. We have clarified that we handle the cumulative incidence in P3L129.
Results, line 143/4: I find “prediction timing of the epidemic size for the season” an unfortunate and confusing choice of words.
>>
We apologize for the confusion. The corresponding sentence was rewritten as: “Vertical lines indicate the week s of each season when the prediction of the epidemic size took place.” (P4L158-L159).
-, lines 145-7: It appears that you, as Goldstein did, chose the thresholds “visually”?
>>
We apologize for the confusion. The thresholds were selected using the optimization with RSE (eq. (2)). Precisions are limited due to sentinel report. We have clarified that thresholds were chosen by equation (2) in P5L161-L162.